# Research on Inter-Turn Short Circuit Fault Diagnosis of Electromechanical Actuator Based on Transfer Learning and VGG16

**Haibin Huangfu [1], Yong Zhou [1,2,*], Jianxin Zhang [3], Shangjun Ma [2,4], Qian Fang [1] and Ye Wang [1]**

[1] School of Aeronautics, Northwestern Polytechnical University, Xi'an 710072, China; huangfuhaibin@mail.nwpu.edu.cn (H.H.); fqxzs1998@mail.nwpu.edu.cn (Q.F.); 2021260213@mail.nwpu.edu.cn (Y.W.)
[2] Xi'an Ding Bai Precision Technology Co., Ltd., Xi'an 710000, China; mashangjun@nwpu.edu.cn
[3] Jiangshan Heavy Industries Research Institute Co., Ltd., Xiangyang 441057, China; xin5137@126.com
[4] School of Mechanical Engineering, Northwestern Polytechnical University, Xi'an 710072, China
* Correspondence: yongstar@nwpu.edu.cn

**Abstract:** In this paper, an inter-turn short-circuit fault of a permanent magnet synchronous motor in an electromechanical actuator is analyzed, and a fault diagnosis method based on transfer learning with a VGG16 convolution network is proposed. First, a 2D finite element model of an inter-turn short circuit fault of a permanent magnet synchronous motor was established in ANSOFT Maxwell, and then a simulation experiment analysis was completed. A three-phase current was chosen as a fault characteristic signal. Second, a fault diagnosis method with a VGG16 deep convolutional neural network and based on transfer learning was designed, and the fine tuning of the hyperparameters of the fault diagnosis model was completed by using grid search and cross verification methods. Finally, based on the transfer learning VGG16 model established in this paper, the inter-turn short circuit fault of a permanent magnetic synchronous machine (PMSM) was diagnosed and verified. The experimental results showed that the proposed convolutional network model based on transfer learning can identify faults effectively and accurately, and has a good engineering guidance significance.

**Keywords:** EMA; PMSM; transfer learning; deep convolutional neural network; fault diagnosis

## 1. Introduction

As electro-mechanical actuators (EMAs) are gradually becoming more widely used in flight control systems, their functioning is closely related to the safety performance of aircraft. In order to prevent failure of the overall flight control system of the aircraft due to the failure of an EMA, and eventually causing serious consequences, it is of great significance to conduct accurate and efficient fault diagnoses for EMA. In the 1960s, researchers focused on fault diagnosis technology, and the original fault diagnosis methods mainly used some simple instruments to gather amplification equipment vibrations or noise signals, and then the relevant experts relied on their own experience to find equipment malfunctions and possible fault sources. This diagnostic method relies heavily on manual work, and it is difficult to form a calibration standard. After the 1970s, with the development of sensor technology and computer technology, the research on fault diagnosis technology gradually entered a new era. The current, vibration, temperature, and other original signals collected by a sensor can be used to extract fault characteristics using the signal processing means, such as spectrum analysis, which improves the reliability of state monitoring and fault diagnosis, and this has become a powerful tool for fault diagnosis. Since the 1990s, intelligent technologies have been emerging, and expert systems of intelligent diagnosis have emerged. They can diagnose automatically, based on their own knowledge and experience. Subsequently, intelligent algorithms, represented by shallow neural networks and support vector machines, have brought important innovations to the field of fault

diagnosis. With the rapid development of deep models in academia and industry, the proposed fault diagnosis method has also achieved great success. Fault diagnosis methods can be divided into the following three categories:

(1)  Model-based fault diagnosis method

Model-based fault diagnosis depends on the cognition of the system, and requires that the fault model is established according to the physical principles, the system model, and the parameters representing the system running state. The signals (or estimated parameters) generated by the model can be used for effective fault detection and identification of possible faults. This type of method requires a clear understanding of the system model and parameter variation range under different operating conditions of the system, because it requires a more in-depth understanding of system research. This kind of method is mostly used by engineers in the control field.

(2)  Signal-based fault diagnosis method

This method does not require building a system model, but depends on the information contained in the signal. The fault signals obtained by modeling or collection are classified into a database, and the operating state or fault situation of the system is obtained by analyzing and processing the various state information in the process of system operation, with a large amount of expert experience. At present, the main signals used are current, vibration, voltage, temperature signal, etc. This approach is based on the behavior of the healthy system, by comparing it with the measured signals. At present, fault diagnosis technologies based on signal processing mainly include EMD, the wavelet transform method, and frequency domain method.

(3)  Data-driven fault diagnosis method

Thanks to the development of artificial intelligence and machine learning, many intelligent diagnosis algorithms have been proposed. These algorithms learn features and intelligently identify fault types, by training using normal data and fault data.

In the existing fault research for EMA, scholars have achieved some excellent scientific research results by using different diagnosis methods.

Balaban et al. developed a health management system for diagnosing and testing EMA faults, which can diagnose EMA faults, track the progress of faults, and predict the remaining service life of actuators using predictive algorithms [1].

Ismai et al. used vibration signal-based fault methods to detect potential EMA faults, without the initial stages of fault feature learning [2]. The two failures considered in the study were high critical blocking and low critical spalling (metal spalling) in an actuator ball screw mechanism. An actuator was used to resample a variable-speed vibration measurement from a single accelerometer at a constant measurement rate. Theoretically, a set of health signatures could be obtained based on the kinematic principle of an EMA ball screw. These theoretical features were compared with those extracted from experimentally measured vibration signals from EMA test products. The vibration signal method was also compared with the diagnostic method, based on EMA motor current measurements. The ability to detect and classify potential failures early on in high-risk or low-risk situations can improve maintenance planning and enhance the reliability of aircraft scheduling.

Berri et al. proposed a data-driven strategy for near-real-time fault detection and isolation (FDI) of dynamic components and the estimation of remaining useful life (RUL) of a system. A scaling Latin hypercube sampling strategy and a new model damage propagation dynamic system method were proposed. This method was evaluated by taking the FDI and RUL estimation of an electromechanical actuator for aircraft secondary flight control as an example. The results showed that this strategy has a high accuracy in the evaluation of system RUL, and is superior to conventional model-based techniques in terms of calculation time [3].

Kawatsu et al. used a model-based fault diagnosis method to conduct relevant research on EMA used in liquid rockets. On the basis of understanding the fault mechanism, the

target object was modeled and simulated under normal and abnormal conditions based on a model, so as to obtain a prior data set for fault diagnosis. Hierarchical clustering was used to classify the data in failure mode [4].

Professor Wang Xinmin of Northwestern Polytechnical University has studied the fault diagnosis of EMA. For various gradual faults of an EMA, sensor signals are collected and high frequency components are removed with a dynamic wavelet. Many faults are detected by using a feedback network method. An interactive multi-model based method was adopted to diagnose an EMA's emergent faults, which improves the accuracy and efficiency of the diagnosis process. In case of multiple faults, this method can detect more serious faults [5,6].

In this paper, the inter-turn short circuit fault of a PMSM, and the core of an EMA, is studied [7,8]. The main work is divided into the following parts: 1. The RMxprt module of ANSOFT Maxwell is used to build a PMSM finite element simulation model for the simulation analysis of an inter-turn short circuit fault of a PMSM. 2. Determine the three-phase current as the fault characteristic signal of the inter-turn short circuit, study the VGG16 network based on ImageNet data set transfer learning, and optimize the basic model. Meanwhile, we discuss several different structures of the top-level model and propose a deep convolutional network based on simulation data [9]. 3. We set up a test bench to simulate inter-turn short circuit faults of different degrees, and compare the network designed in this paper with the traditional VGG16 network, to complete the verification of fault diagnosis [10,11]. The experimental results showed that compared with the traditional VGG16 network, the method proposed in this paper needed less training time, simplified the number of network parameters, reduced computing requirements, and is suitable for more hardware environments. The method proposed in this paper has proven to be effective and will work better for engineering practice.

The characteristics and innovation of this study are mainly reflected in the following: a data-driven PMSM fault diagnosis method is established, a fault diagnosis model based on transfer learning and VGG16 neural network under ImageNet data set is proposed, and the simulation data with Gaussian noise were used to train a network model. After the training, the fault experimental data collected were used to verify the method, and the method was compared with the VGG16 network model, without changing the network structure and without transfer learning. The experimental results showed that the VGG16 network based on transfer learning proposed in this paper has almost the same accuracy as the VGG16 network, without changing the network structure and without transfer learning, but the training time was only 53.9% of VGG16, which can greatly save network training time and hardware costs, on the basis of ensuring the accuracy of fault diagnosis. This method has strong engineering application value.

## 2. Motor Model

### 2.1. Finite Element Model of a Motor

When a fault occurs in the PMSM, its phase self-inductance and mutual inductance will change significantly; because the calculation error of the mathematical model is too large to accurately describe the motor, and the finite element method has good analytical and computational capabilities, we can easily solve the parameter changes when the fault occurs by setting different types of faults in the finite element software.

Maxwell is widely used in electromagnetic design and the simulation of motors. Due to its powerful ability in electromagnetic simulation, this paper uses this software to establish an electromagnetic transient analysis model of a motor, and a conduct simulation of inter-turn short circuit fault model.

When a turn-to-turn short circuit occurs in a PMSM, the permanent magnets will lose excitation at the same time. The reason for this is that the magnetic fields generated by the short-circuit coils are in opposite directions. The causes can be summarized as follows: (1) high temperature of the PMSM stator winding; (2) vibration of the PMSM; (3) overvoltage when the PMSM starts up; and (4) long-term harsh working environment of the PMSM. A

turn-to-turn short circuit will lead to an unbalanced impedance of the three-phase windings of a PMSM, resulting in an unbalanced current, which will make the PMSM vibrate.

In this paper, the RMxprt module in ANSOFT Maxwell was used to build a fault model of an inter-turn short circuit of a motor, and based on this model, simulation of an inter-turn short circuit was completed. First, the type of motor was selected. There are 13 types of motors in the general template library. The line-start permanent magnet synchronous motor was selected for PMSM modeling.

Maxwell requires setting the parameters of the machine, stator, rotor, and shaft of the motor. The specific settings were as follows:

(1) Machine was set to two pairs of poles, the given speed was 1500 rpm, and the friction loss corresponding to this given speed was 160 W;

(2) Stator was set with an outer diameter of 260 mm, inner diameter of 170 mm, length of stator core of 190 mm, material of DW315_50, effective magnetic length coefficient in superposition factor field with the default value of 0.95, and number of stator slots of 36; the stator slot type was the pear-shaped slot; the number of stator winding layers was 1, the number of conductors per slot was 13, and the number of turns per slot of single-layer winding was the number of conductors per slot; so the number of turns per slot was 13, and the number of wires per turn was 5. Figure 1 shows the three-phase winding arrangement and distribution of the PMSM:

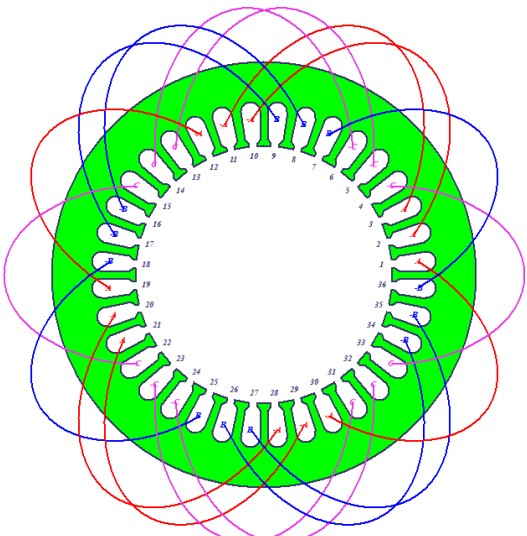

**Figure 1.** Arrangement and distribution diagram of three-phase windings.

(3) The rotor was set with outer diameter of 168.7 mm, inner diameter of 60 mm, length of rotor core of 190 mm, material of DW315_50, number of rotor slots of 32, effective magnetic length coefficient in superposition factor field as the default value of 0.95, and with the surface-mounted magnetic pole type.

Then, the Analysis-Setup was set, and the PMSM used a constant power load, with a rated output power of 15 kW, a rated voltage of 380 V, and a rated speed of 1500 rpm.

As above, the RMxprt motor parameter setting was completed, and the correctness of the motor model was tested. When there was no error, the equivalent circuit and magnetic circuit method were used to solve it, and the RMxprt solution result of the PMSM was obtained. After the solution was completed, it could be converted into a 2D model of the motor using the 'Create Maxwell Design' conversion module, and the other steps could be automatically completed during the solution process. A two-dimensional model of the motor is shown in Figure 2:

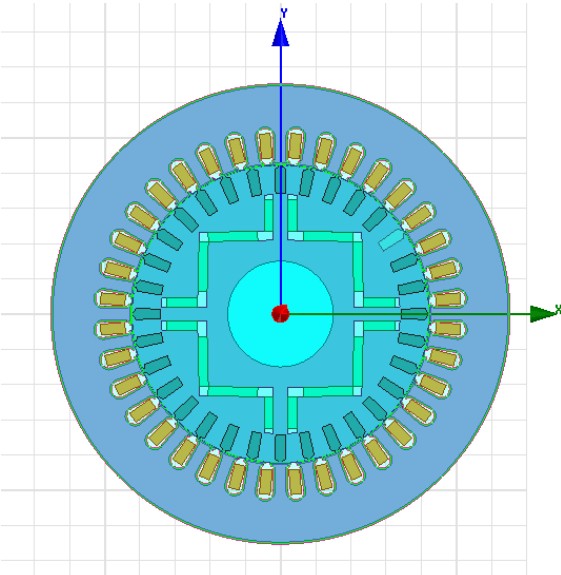

**Figure 2.** 2D finite element model of motor.

Before transient analysis, it was necessary to set simple parameters of the motor. The solution setting setup1 was kept unchanged when RMxprt was generated, and we changed the simulation time to 0.2 s. At the same time, the MotionSetup1 of the motor should be set, specifically as follows: the initial angular velocity was 0; moment of inertia and damping were automatically calculated according to the setting of motor RMxprt; and the torque was set to 0. After setting these parameters, the model needs to be tested with the finite element method and the transient field can be solved after the test is completed.

After the solution was completed, the solution results of the transient field could be derived from the results module. The transient field result-natural coordinate system was created, and then the torque, speed, three-phase current or voltage of the stator winding were selected to obtain the simulation diagram of the transient field in this case.

### 2.2. Influence of Short Circuit on the Three-Phase Current of the Motor

For a turn-to-turn short circuit fault of a certain phase of a PMSM stator winding, 1%, 2%, 4%, 8%, and 16% of turn-to-turn short circuit faults were injected, respectively, by modifying the coil turns of the A-phase branch I of the stator three-phase winding. The windings of the first branch of the A-phase were PhA_0, PhA_1, PhA_2, PhARe_0, PhARe_1, and PhARe_2. Figure 3a–f shows the three-phase current curves of the motor under different conditions. In Figure 3a,b, there are two current waveforms in the normal state and 1% turn-to-turn short circuit state. There is no obvious difference between the three-phase current, and the average and variance of the three-phase current. In the state of a 2% turn-to-turn short circuit in Figure 3c, compared with the first two, at this time, the phase B current is slightly reduced compared with A and C, and the mean value and variance of the three-phase current are different. Under the condition of a 4% turn-to-turn short circuit in Figure 3d, compared with the previous ones, the current of A and C increases more obviously, and the phase current of B decreases more. In the state of an 8% turn-to-turn short circuit in Figure 3e, compared with the previous ones, the current of A and C increases more obviously, and the phase current of B decreases more. In the state of a 16% turn-to-turn short circuit in Figure 3f, compared with the previous ones, the current of A and C increases to the maximum, and the phase current of B is reduced to the minimum value. It can be inferred that with the increase of the turn-to-turn short circuit degree of phase A, the peak values of phase A and C will gradually increase, and the phase B will gradually decrease.

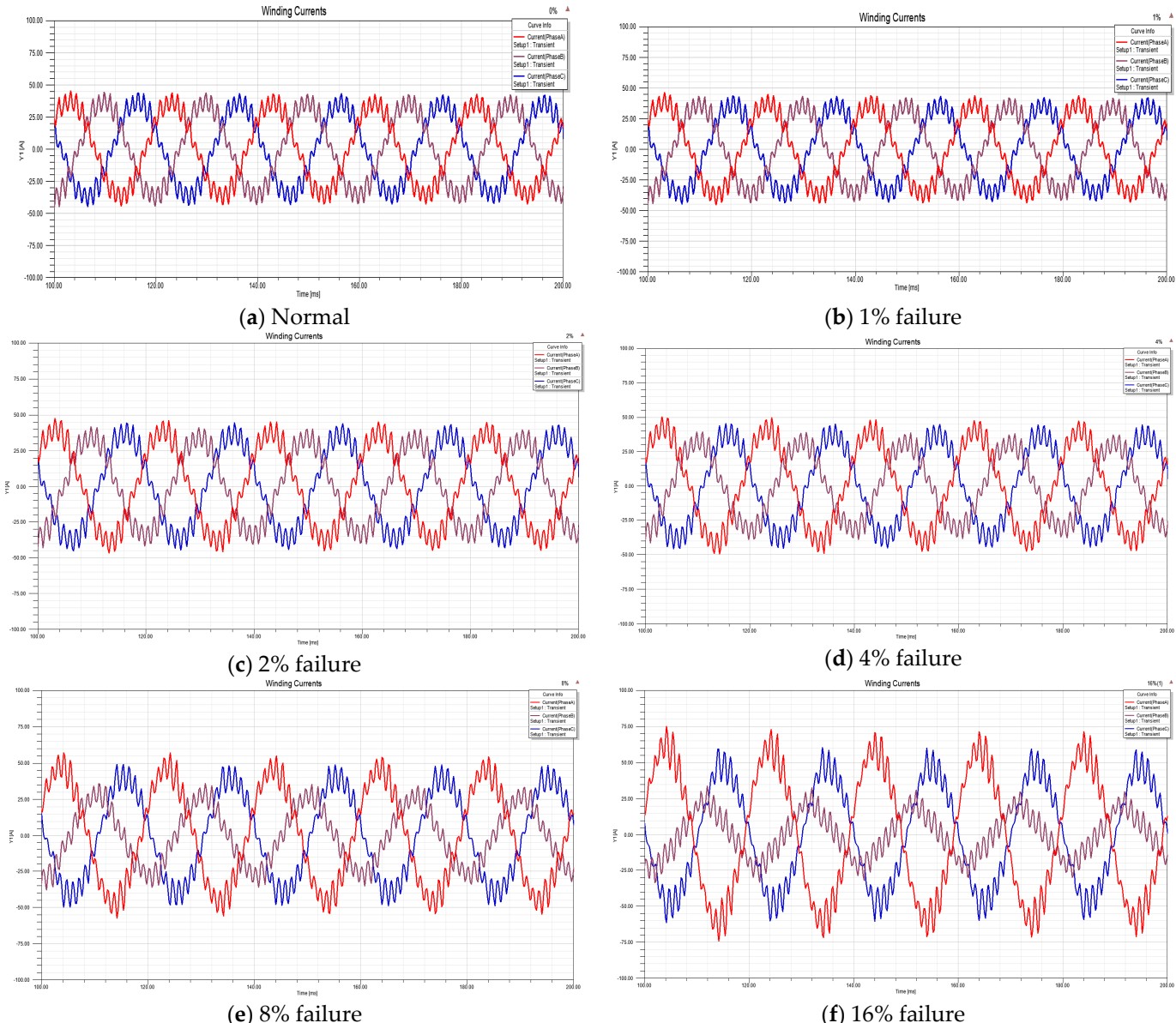

**Figure 3.** Three−phase Current Curve of Motor under Different Conditions.

When the motor is normal, the three-phase current value is about 43 A. When a fault occurs, the change trend of the three-phase current is as shown in Figure 4. It can be seen from Figure 4 that, with the increase of fault degree, the phase current values of phase A and C gradually increase, and the phase current values of phase B gradually decrease. When the turn-to-turn short circuit fault of phase A is 16%, the maximum values of phase A, phase C, and phase B are 71.4 A, 60.3 A, and 30.1 A, respectively.

The three-phase current data obtained by simulation were used as a training sample set. The sample set was an array with $3 \times 50{,}000$ three-phase currents removed from the time axis, with every 400 points as a cycle, and with $3 \times 32 \times 32$ points as a sample, and the sample set was expanded using the data enhancement method. Data enhancement mainly includes the following methods: for picture data, on the one hand, we can rotate a certain angle, flip, zoom, and other methods can be used; on the other hand, we can use color transform, noise, fill, etc. For the three-phase current data in this paper, the method adopted was similar to the vibration signal data increment method of rolling bearings, as shown in Figure 5. When dividing the samples, we used overlapping data with a certain

step size; that is, more sample data could be obtained. In this way, the sample size could be expanded to 5000 samples, and the sample set could be expanded 100 times.

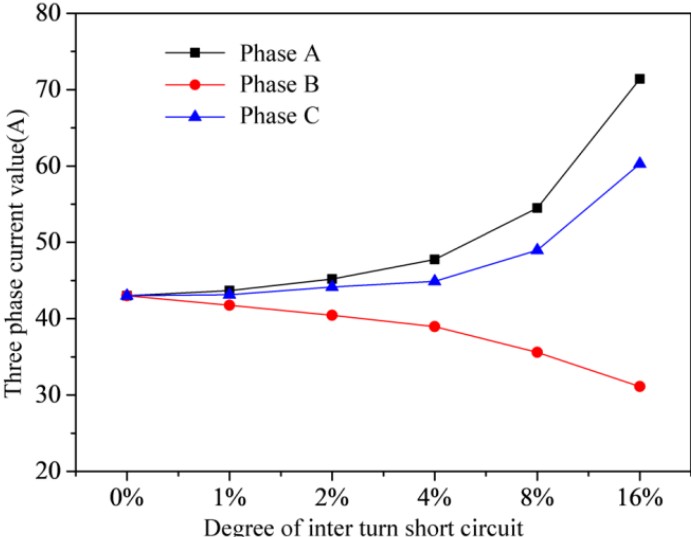

**Figure 4.** Trend of three−phase current.

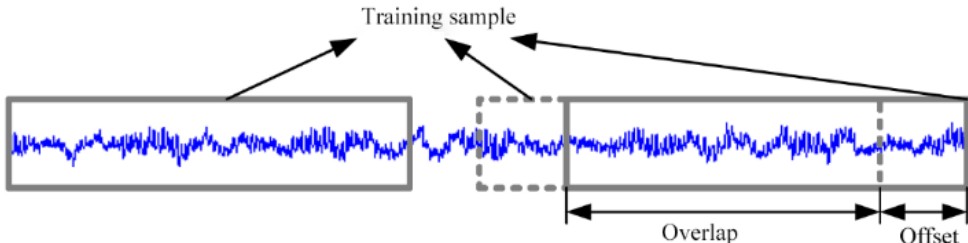

**Figure 5.** Data enhancement method.

For the sample set data, Gaussian white noise was superimposed using MATLAB, and the current signal was closer to the real three-phase current signal after adding the noise. By adding the noise, this section becomes more valuable for discussing the network structure and parameters.

## 3. Fault Diagnosis Method

### 3.1. Research on Network Structure

The VGG16 network, proposed in 2014, has good transfer learning ability and generalization ability, which can further reduce the time cost of grid training. The input of VGG16 is an RGB image of a certain size, followed by a series of blocks. Each block is divided into continuous stacked convolution layer and max pooling layer. After output from five consecutive units, including block1–block5, three layers of full connection (FC) are connected, whose activation function is ReLU. After each layer, we connected dropout to prevent network overfitting and merge to enhance the robustness of the model. The last layer was a SoftMax layer. Figure 6 shows the network structure of the VGG16; the whole network model contains 138,357,544 parameters. In Keras, a deep learning framework with TensorFlow as the back end, the weight of the pre-training network model could be directly learned by transferring ImageNet data sets.

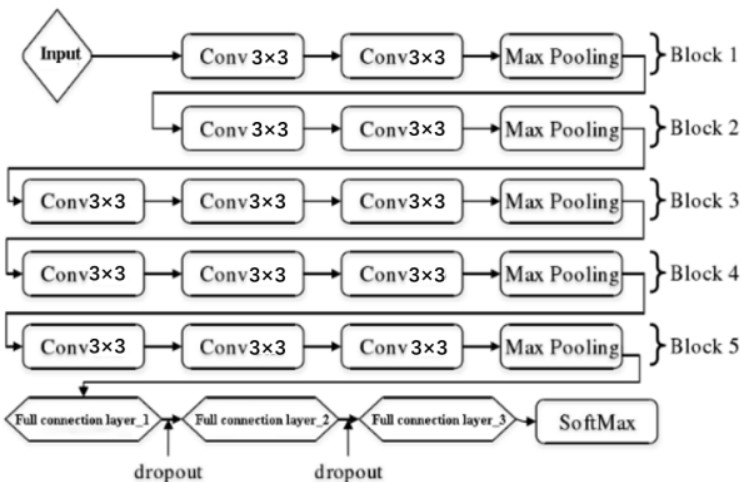

**Figure 6.** VGG16 structure diagram.

The method of this paper is based on the VGG16 model of ImageNet data set and transfer learning; retaining the first five blocks, removing the top three full connection layers, and freezing the pre-trained weight values. The parameters of the top model were unified as follows: (1) the cross entropy function was selected as the loss function of the top model; (2) the momentum random gradient descent method was selected to optimize the model, and the learning rate was set to 0.0001, and the momentum was set to 0.9; (3) batch-size was the default value of 32; and (4) epochs was set to 30.

For transfer learning, when the new sample set is small and quite different from the original sample set of transfer learning, choosing fewer layers and a linear classifier will usually produce better results. The network structure discussed in this paper is shown in Table 1. The network structure in the table is A, B, C, D, E, F, G, and H from top to bottom:

**Table 1.** Network Structure.

| Basic Model | Top Model | Category |
|---|---|---|
| Block1~5 Freeze pre-training parameters. | Three-layer fully connected layer with no activation function + SoftMax layer. | A |
| | Three-layer fully connected layer has activation function + SoftMax layer. | B |
| | Two fully connected layers with no activation function + SoftMax layer. | C |
| | Two fully connected layers with activation function + SoftMax layer. | D |
| Block1~4 Freeze pre-training parameters. | Three-layer fully connected layer with no activation function + SoftMax layer. | E |
| | The three-layer fully connected layer with activation function + SoftMax layer. | F |
| | Two fully connected layers with no activation function + SoftMax layer. | G |
| | Two fully connected layers with activation function + SoftMax layer. | H |

The accuracy of the verification set of fault diagnoses was about 90% in most network structures, and the advantages of the transfer learning network were fully demonstrated. In the process of 30 iterations, the G-basic model of network structure was Block 1~4 freezing pre-training parameters, and the top model was two layers of fully connected layer with no activation function + SoftMax layer, with the highest accuracy. This shows that the training effect was better with fewer fully connected layers and linear models that do not use the activation function in small sample transfer learning [12,13].

Regarding the change of cross entropy loss of the verification set, as shown in Figure 7, Figure 7a shows the structures A, B, C and D; and Figure 7b shows E, F, G, and H. This basically corresponds to the trend of the accuracy. Due to the pre-training weight, the value of the loss after the first iteration was relatively small. Compared with other top-level structures, network structure G has obvious advantages.

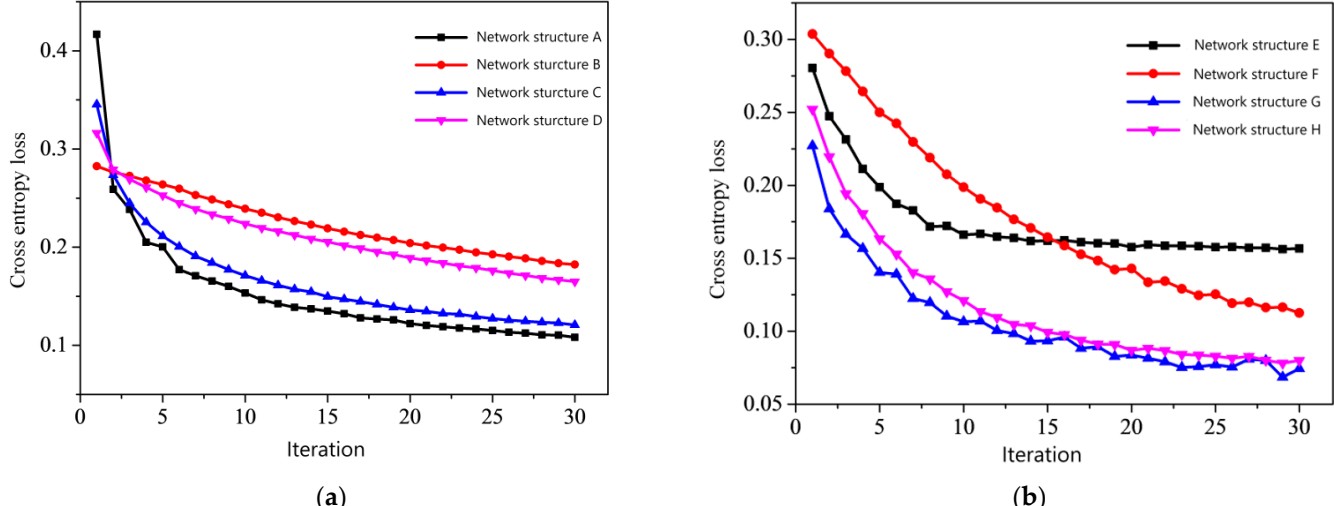

**Figure 7.** Change of cross entropy loss in the fault diagnosis of each network. (**a**) Shows the structures A, B, C and D, (**b**) Shows the structures E, F, G, and H.

After training each network 30 times, the training effect was tested on the test set. Details of the test set are shown in Figure 8a,b. The results of the test set were similar to those in the training process. The network structure G had the lowest cross entropy loss and the highest accuracy in the test set.

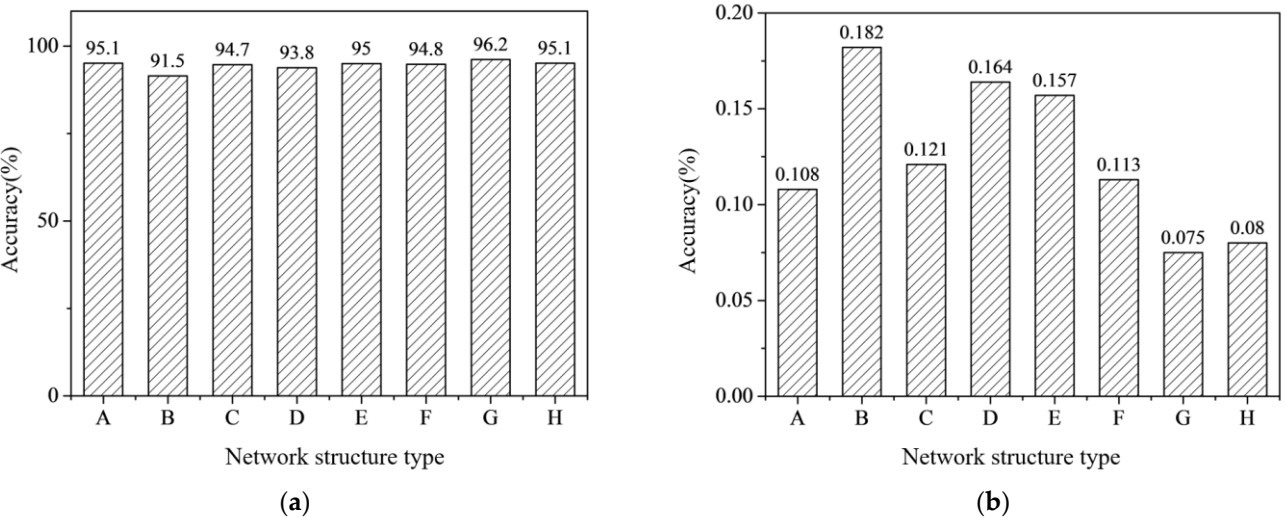

**Figure 8.** Accuracy comparison and cross-entropy loss comparison of each network structure model. (**a**) Accuracy comparison; (**b**) Cross entropy loss comparison.

The time cost of network training is also an important factor to be considered, and the network structure greatly affects the training time of a network. As shown in Figure 9, compared with these different network structures, the training time of freezing Blocks 1~4 was slightly longer than that of freezing Blocks 1~5, which was about 16% longer on average. The training time of network structure G was slightly inferior. On the whole, network structure G had the best performance [14,15].



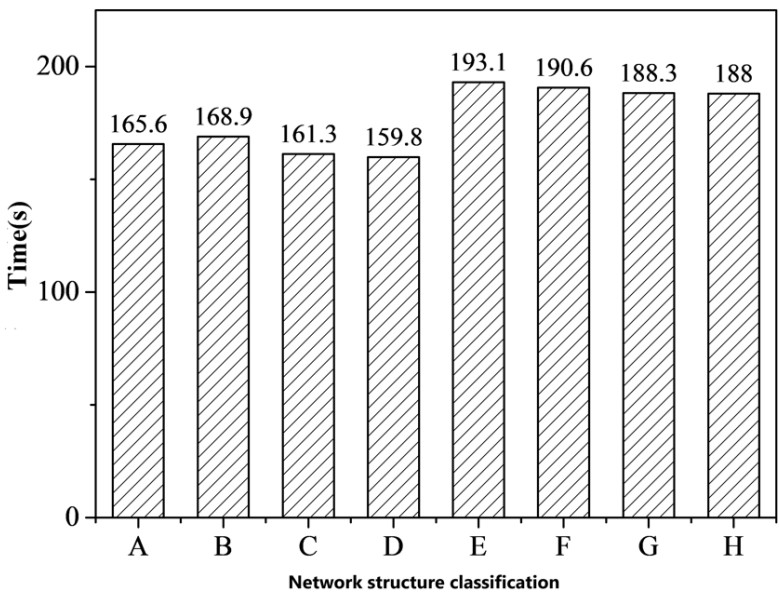

**Figure 9.** Time spent by each network after 30 trainings.

Therefore, in this paper, the basic model includes Block 1~4 freezing pre-training parameters, and the top model is a two-layer fully connected layer with no activation function + SoftMax layer structure, as the fault diagnosis method.

### 3.2. Fine Tuning of Hyper-Parameters

In this paper, the grid search and cross-validation (GridSearchCV) of Scikit-Learn was used to complete the searching of hyper-parameters. The hyper-parameters that greatly affect this network are batch-size and learning-rate. The candidate values of batch-size were 4, 8, 16, 32, 64, 128, and 256, and the candidate values of learning-rate are 0.1, 0.01, 0.001, 0.0001, and 0.0001. Figure 10 is the heat map after GridSearchCV, with the abscissa being learning-rate and the ordinate being batch-size, while the color from dark to light indicates the average score of ten times, from low to high. GridSearchCV had the highest automatic output score, which corresponded to a learning-rate of 0.01 and a batch-size of 64.

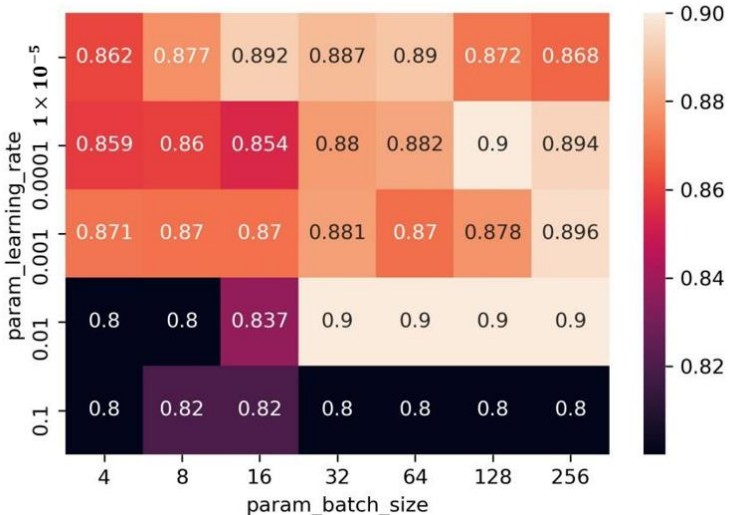

**Figure 10.** Network score heat diagram of each group with hyper-parameters.

Therefore, the fine tuning results of network hyper-parameters in this paper were that the learning-rate was 0.01, and the batch-size was 64.

## 4. Experiment and Fault Diagnosis Verification Motor Model

### 4.1. Hardware Design of Experimental Platform

The experimental platform was mainly composed of the following parts: upper computer, electromechanical actuator and its controller, sensor and data acquisition system, and loading system and power supply system. The overall structure is shown in Figure 11.

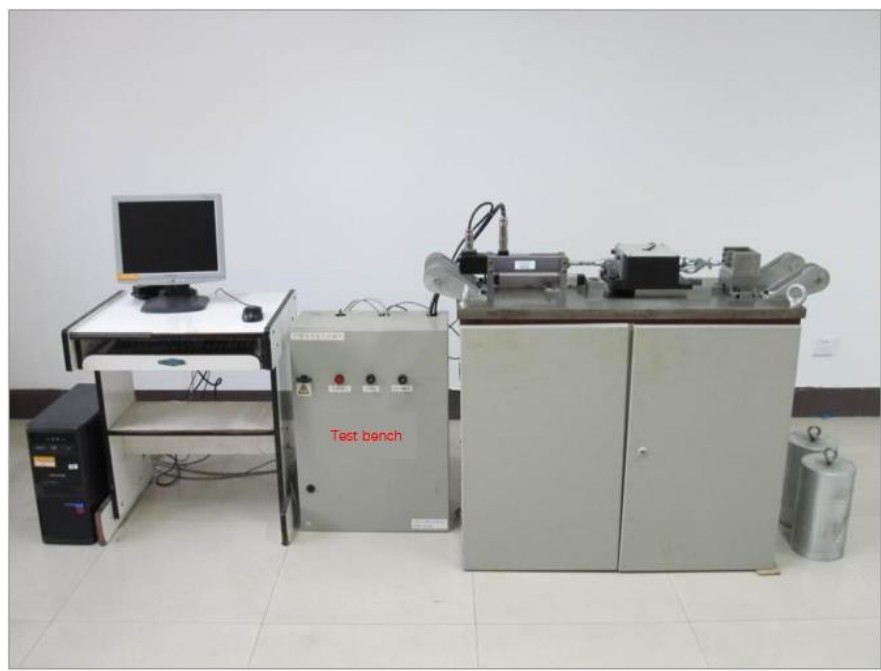

**Figure 11.** Overall structural diagram of the experimental platform.

### 4.2. Software Design of the Experimental Platform

The fault experiment platform software was written and developed by LabVIEW, which is mainly composed of the following parts: initialization module, signal acquisition module, waveform display module, data storage module, protection module, and PI control module.

### 4.3. System Test and Fault Diagnosis Verification

Based on the hardware and software design of the fault experiment platform, this section evaluated different degrees of turn-to-turn short-circuit faults. After the collected three-phase current data were processed, the method proposed in this paper was compared with VGG16, and the fault diagnosis and verification were completed

#### 4.3.1. System Testing

Each phase winding of a PMSM can be simplified as the inductance connected in series with an electric group, ignoring the influence of inductance, and then each phase is Rs. Connect R between the controller and the three-phase winding of a PMSM in series, and take this Rs + R as the new internal resistance of each phase of the PMSM. As shown in Figure 12:

The specific method of series connection is that the three-phase windings of PMSM are, respectively, connected in series with resistors (one 0.5 Ω, two 0.2 Ω, one 0.1 Ω) with a total resistance of 1 Ω, which are used as the benchmark of the motor to simulate the normal working state of the PMSM. Figure 13 shows the current waveform collected in the normal state, where the peak value of the three-phase current is about 1.4 A. Due to the interference of the noise signal, the collected current signal is greatly distorted. At this time, the actuating system works normally and the output speed is stable.

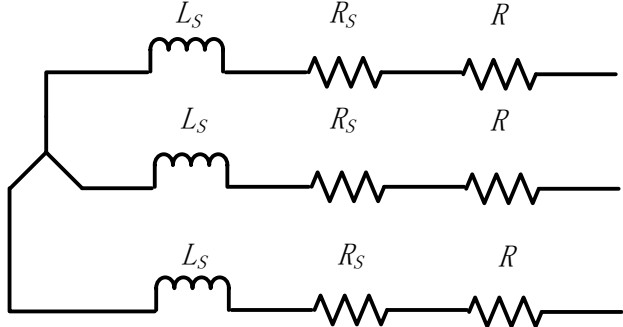

**Figure 12.** Three-phase winding fault injection.

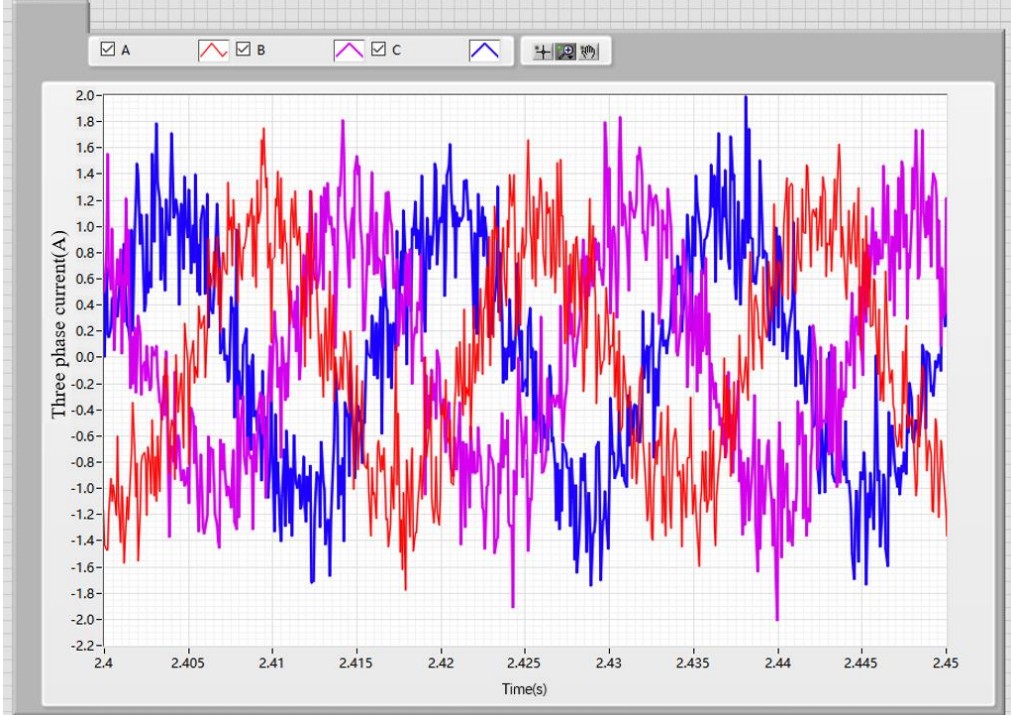

**Figure 13.** Sampling waveform of a three−phase current of motor in normal state.

To simulate a slight turn-to-turn short circuit of a PMSM, we removed 0.1 Ω resistance of Phase A. As shown in Figure 14, the current waveform in the state of slight inter-turn short circuit is shown. When the fault is minor, there is no significant change in the three-phase current of the motor.

For simulating a moderate turn-to-turn short circuit in a PMSM, 0.2 Ω resistance was removed from phase A. As shown in Figure 15, the current waveform of the medium turn-to-turn short circuit state was collected. During the medium fault, the change trend of the three-phase current of the motor was similar to that of the simulation, with phase A and phase B increasing and phase C decreasing.

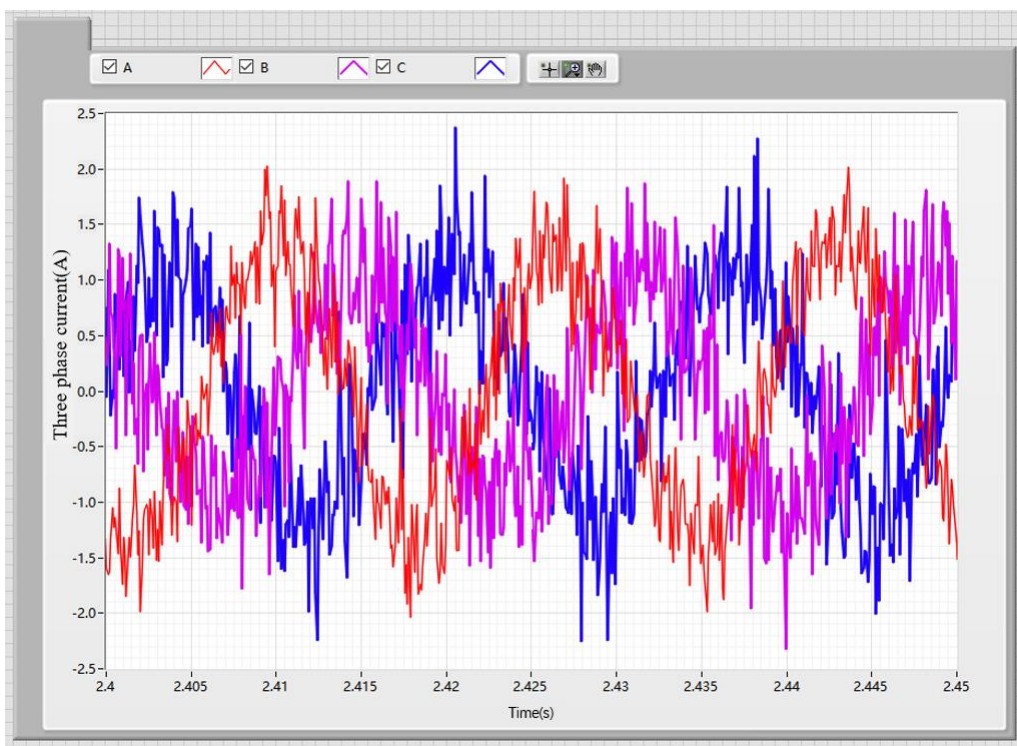

**Figure 14.** Sampling waveform of a motor three−phase current in a mild turn−to−turn short circuit state.

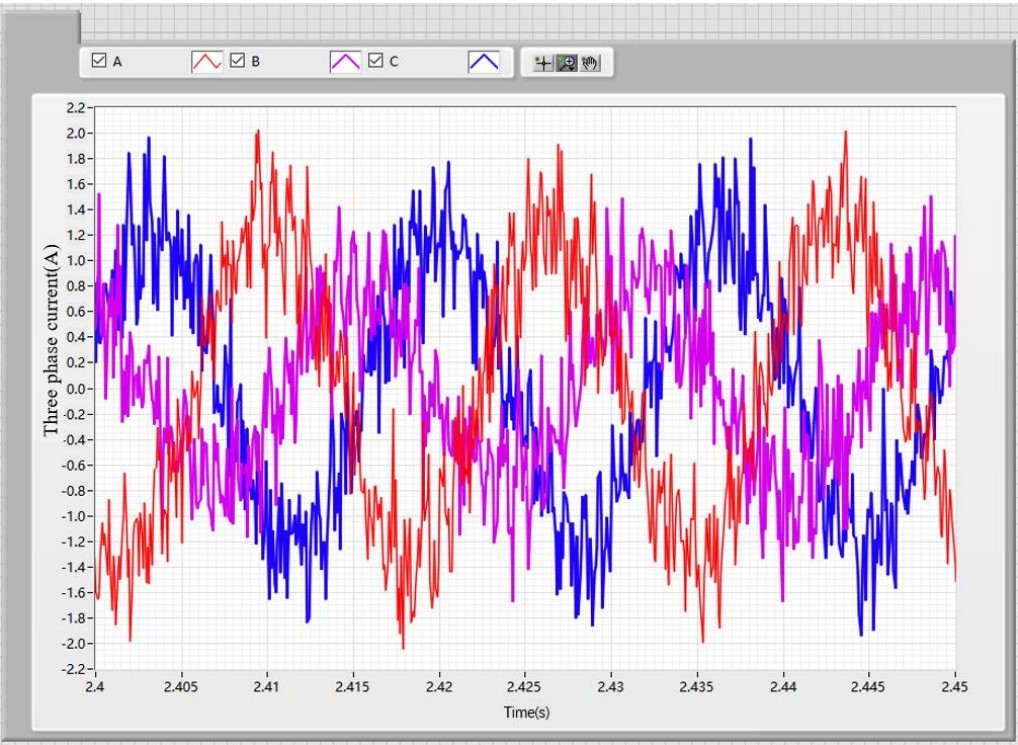

**Figure 15.** Three−phase current sampling waveform of a motor in a medium turn−to−turn short circuit state.

For simulating a severe turn-to-turn short circuit of a PMSM, two 0.2 ω resistors were removed phase A. As shown in Figure 16, the current waveform of a severe turn-to-turn

short circuit state was collected. With a severe fault, phase A and phase B currents increased more, while phase C currents decreased more.

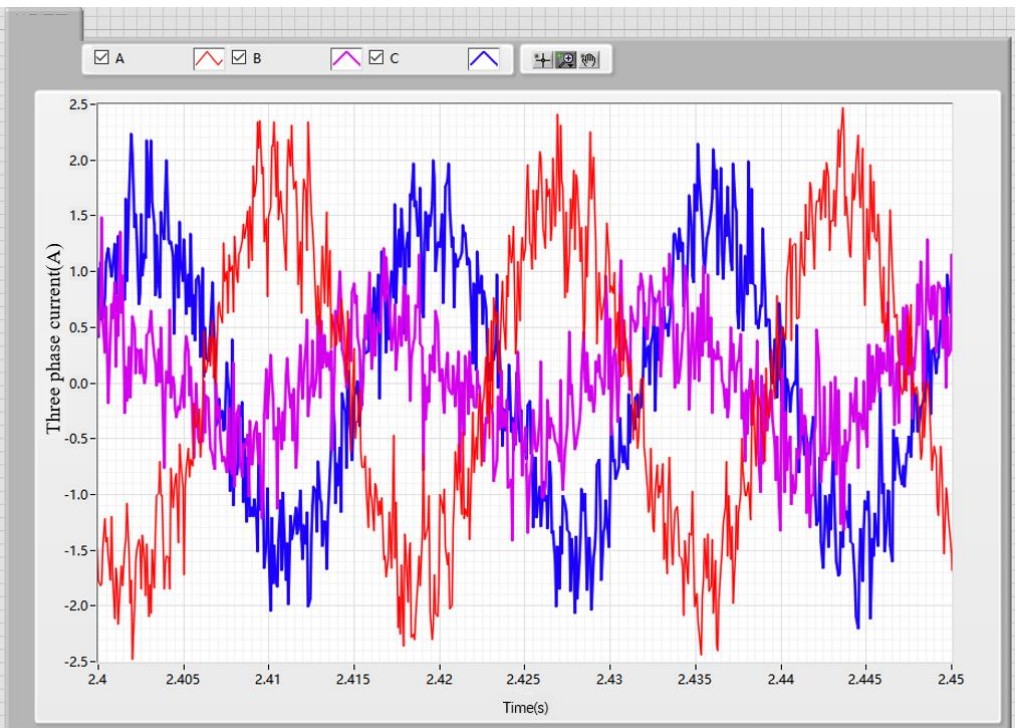

**Figure 16.** Sampling waveform of a motor three−phase current in a severe turn−to−turn short circuit state.

### 4.3.2. Fault Diagnosis Verification

For the sampled three-phase current data of a PMSM in the four states, similar operations were performed on the experimental data and simulation data. A total of 32 × 32 sampling points of each phase of the three-phase current were taken as a sample, so each sample was 32 × 32 × 3. The sampled data were divided into 1000 samples, and divided into a 60% training set, 20% test set, and 20% verification set and labeled with one-Hot coding.

In this paper, the fault diagnosis model is the structure established in Section 3. The basic model is the Block 1~4 freezing pretraining parameters of VGG16, and the top model is the structure of the two-layer fully connected layer without activation function + Softmax layer. The related parameters were consistent with those described above.

The method compared with that in this paper was the VGG16 deep convolutional network, which does not change the network structure and does not load the weight value of transfer learning. The hyperparameters of this network are default values.

Figure 17 shows the prediction results after 30 iterations. By comparing the two methods, it can be seen that the prediction accuracy of the two methods was 96.1% and 96.7%, respectively, which are very close and at a similar level. However, in terms of the network training time, the method in this paper greatly accelerated the training speed and reduced the network training time, due to the effect of the frozen pre-training weight of transfer learning, and the training time was only 53.9% of VGG16. In conclusion, the method based on transfer learning and VGG16 proposed in this paper can greatly reduce the training time of the network on the basis of ensuring the accuracy of fault diagnosis, and has a strong engineering application value.

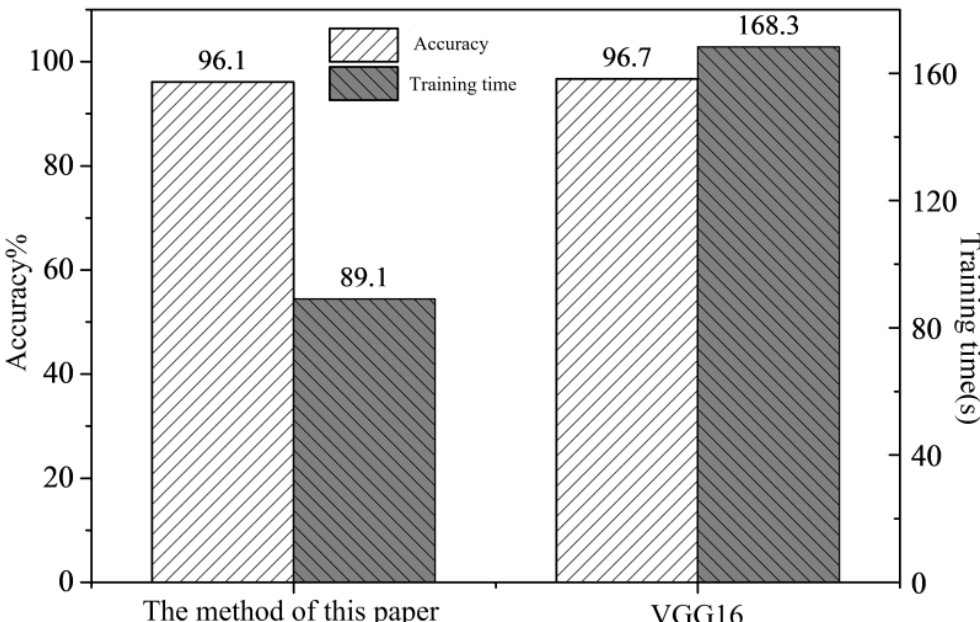

**Figure 17.** Comparison of the two methods.

## 5. Conclusions

With the development of all-electric aircraft and the wide application of electromechanical actuators, many researchers have started to research the fault diagnosis of electromechanical actuators, which is of great significance for preventing the electromechanical actuator systems from overall failure, due to electrical failure. This paper mainly studied the important electrical faults of a permanent magnet synchronous motor in an electromechanical actuation system. As the core of an electromechanical actuation system, the faults of the motor greatly affect the working state of the whole system. In this paper, the fault modes of electromechanical actuators were explored, and the typical faults of permanent magnet synchronous motor were modeled and analyzed in ANSOFT Maxwell. A deep convolution neural network model based on transfer learning and VGG16 was designed, and a fault simulation and signal acquisition test bed was built. The fault diagnosis method proposed in this paper was used to verify the diagnosis effect. The main research results of this paper include:

1. The 2D finite element model of a turn-to-turn short circuit fault of a permanent magnet synchronous motor was established in ANSOFT Maxwell, and a simulation experiment analysis was completed. On the basis of the experimental analysis, a three-phase current was determined as the fault feature.

2. Based on transfer learning and VGG16, a new fault diagnosis model was built in this paper. Due to the important role of transfer learning in small sample data sets and the advantages of a deep convolution network in fault diagnosis, a fault diagnosis method with a VGG16 deep convolution neural network based on transfer learning with the ImageNet data set was proposed. The simulation data were used as the sample set, and the data of the sample set were enhanced by changing the segmentation method, which increased the number of samples. At the same time, Gaussian white noise was added to the sample set data to make the simulation data closer to the real data. The influence of network structure on fault diagnosis was studied. Based on VGG16, several different structures were designed, and the whole network structure was determined. Grid search and cross-validation were used to fine-tune the hyperparameters Batch-size and Learning-rate and to further optimize the method proposed in this paper.

3. We designed and built a fault simulation and signal acquisition test bed. The software design was based on LabView, and different software modules were completed according to different requirements, and different turn-to-turn short circuit faults were

designed and simulated. The hardware design includes an upper computer for the electromechanical actuator system, the electromechanical actuator and its controller, the sensor and data acquisition system, the loading system, and the power supply system. By connecting high power and low resistance in series, inter-turn short circuit faults of a PMSM of different degrees were simulated.

4. The verification of the fault diagnosis method was completed. Three-phase current data of a EMA fault state were collected using test-bed. The method was compared with the VGG16 network without transfer learning, the test data were diagnosed. The experimental results showed that the method proposed in this paper can accurately diagnose different degrees of turn-to-turn short circuit faults. Compared with VG16, the diagnostic accuracy was close to the VG16, and the training time was only 53.9% of that of VG16. This proved that the method proposed in this paper can complete fault diagnosis effectively and quickly, and has a good engineering application value.

## 6. Follow up Work and Prospect

The fault diagnosis model described in this paper can be used for offline permanent magnet synchronous motor fault diagnosis, and has a high accuracy, as well as needing less training time and saving hardware costs. In addition, due to the limitations of experience and research conditions, the research work in this paper still needs further exploration:

1. For the fault diagnosis algorithm, only the two most important hyperparameters were considered in the hyperparameter fine-tuning, and more hyperparameters should be fine-tuned in the follow-up, so as to better optimize the diagnosis model and improve the accuracy of fault diagnosis.

2. The method proposed in this paper is an offline fault diagnosis. Online prediction is more meaningful in engineering practice. Therefore, online application research of diagnostic models is the most important direction for improving engineering practice.

**Author Contributions:** H.H., Y.Z., Q.F., Y.W. were in charge of the whole trial; H.H. wrote the manuscript; J.Z. and S.M. assisted with sampling and laboratory analyses. All authors have read and agreed to the published version of the manuscript.

**Funding:** The research was supported by Key R & D projects in Shaanxi Province (2021ZDLGY10-08) and the National Natural Science Foundation of China (Grant No. 51875458, 51905428).

**Institutional Review Board Statement:** Not applicable.

**Informed Consent Statement:** Not applicable.

**Data Availability Statement:** Data sharing is not applicable.

**Conflicts of Interest:** The authors declare no conflict of interest.

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
