# Peer review of "Research on Inter-Turn Short Circuit Fault Diagnosis of Electromechanical Actuator Based on Transfer Learning and VGG16"

_electronics, doi:10.3390/electronics11081232_

Round 1

Reviewer 1 Report

General Comments: In this paper, the inter-turn short-circuit fault of permanent magnet synchronous motor in electromechanical actuator is analyzed, and a fault diagnosis method based on transfer learning of VGG16 convolution network is proposed. First of all, the 2D finite element model of inter-turn short circuit fault of permanent magnet synchronous motor was established in ANSOFT Maxwell and the simulation experiment analysis was completed. It was decided to use three-phase current as fault characteristic signal. Secondly, a fault diagnosis method of VGG16 deep convolutional neural network based on transfer learning is designed, and the fine tuning of the hyperparameters of the fault diagnosis model is completed by using grid search and cross verification methods. The authors are advised to do the following important in the revised paper for the better understanding of the readers

  1. Avoid the abbreviation in the abstract section. Application of the current inspection should be included in the abstract section
  2. Provide the appropriate reference for table 1. Why finite element method is used? 
  3. Explain the graphical results in detail
  4. Extend the conclusion section. Improve the introduction section of the draft
  5. the novelty of the work should be added in the last paragraph of introduction section
  6. authors should add the future recommendation and application of their work for the betterment of society

Reviewer 2 Report

In this paper, the inter-turn short-circuit fault of permanent magnet synchronous motor 15 in electromechanical actuator is analyzed, and a fault diagnosis method based on transfer learning 16 of VGG16 convolution network is proposed.

Comments:

The quality of the figures is low. For example: Fig. 3, 5, 9, 10, 11, 12,

Please compare your work with others, and report in a table.

Please explain more about figure 2 (2D finite element model)

If it is possible add experimental results.

I suggest authors to do an English proofreading by a native speaker. It would improve the quality of the paper.

Author Response

This manuscript is a resubmission of an earlier submission. The following is a list of the peer review reports and author responses from that submission.